# Economic policy uncertainty and common prosperity within the enterprise: Evidence from the Chinese market

**Linjing Yang, Xiaoke Tan**[ID]**\*, Guifang Tan**

The Forge Business School, Chongqing college of Mobile Communication, HeChuan, Chongqing, China

\* xiaoke.tan.461875@hotmail.com

**Data Availability Statement:** All relevant data are within the paper and its Supporting Information files.

**Funding:** This work was supported by a grant from the Chongging Municipal Education Commission

## Abstract

In the continuous development of the Chinese economy, the desire for common prosperity among the people has become increasingly strong. However, the complex domestic and international situations, along with the economic uncertainties post-pandemic, have introduced significant uncertainties into government economic policies. In the long term, this uncertainty has a profound impact on intra-enterprise common prosperity, affecting society at large. This paper focuses on non-financial listed companies on the Shanghai and Shenzhen stock exchanges from 2011 to 2020, examining the negative effects of economic policy uncertainty on intra-enterprise common prosperity. The results indicate that economic policy uncertainty has a significant adverse impact on intra-enterprise common prosperity, and this impact is more pronounced in state-owned enterprises, those in the growth life cycle, and companies with poor innovation capabilities and limited external support. Companies enhancing social responsibility and overall factor productivity can help mitigate these negative effects. Furthermore, in dynamic features, as the level of intra-enterprise common prosperity increases, the negative impact of economic policy uncertainty gradually diminishes, while the positive effects of overall factor productivity and corporate social responsibility on enterprises become more evident.

## 1. Introduction

Achieving common prosperity is the core goal of socialist modernization and a strategic requirement of the Party and the country. The goals for 2035 and 2050 proposed in the 19th National Congress Report include raising people's income levels, narrowing income gaps, improving the public service system and achieving common prosperity, which marks that common prosperity has risen to a national strategy. In June 2021, Zhejiang Province was designated as a demonstration zone for common prosperity, emphasizing that common prosperity is the only way to achieve a happy life through hard work and sharing of development results.

Common prosperity within enterprises is one of the important ways to achieve social common prosperity. In the market mechanism, enterprises, as basic economic entities, have

under Grant No. 24SKGH370. The funders had no role in study design, data collection and analysis, decision to publish, or preparation of the manuscript.

**Competing interests:** The authors have declared that no competing interests exist.

production and business activities that directly affect the creation and distribution of wealth. Common prosperity within enterprises refers to the fair distribution and growth of income within the company, and various production factors (such as capital, labor, management and technology) share the results of operations according to their contribution levels. Common prosperity within enterprises promotes harmonious relations among corporate stakeholders and creates a good internal and external environment for sustainable development [1]. However, some scholars have found that the operating conditions of enterprises depend not only on their own capabilities and competitiveness, but also on the government's economic policies [2]. Considering the current trade protectionism, weak international market, and epidemic disturbances, the uncertainty of the external economic and policy environment faced by enterprises has increased significantly. How enterprises cope with the impact of external economic policy uncertainty to achieve common prosperity within the enterprise has important theoretical and practical significance for understanding and promoting the realization of the common prosperity process.

Economic policy uncertainty refers to the probability of economic policy changes and fluctuations caused by the uncertainty of economic policy makers and the complexity of the decision-making process. Economic policy uncertainty will affect the company's income level, income structure, production decisions and investment decisions, and then affect the company's internal income distribution strategy [3]. Economic policy uncertainty also increases information and transaction costs, reduces investment, consumption, innovation and employment willingness, and affects economic growth and stability [4], thereby affecting internal income distribution [5]and the allocation and utilization of production factors, further affecting innovation motivation, technological progress and efficiency improvement [6], and ultimately affecting the company's sustainable development capabilities and social responsibility. In summary, total factor productivity and corporate social responsibility play an important role in enterprises coping with economic policy uncertainty to achieve common prosperity within the enterprise.

This paper uses panel data of Chinese A-share listed companies from 2011 to 2020 to study the impact of economic policy uncertainty on internal prosperity of enterprises, and explores the heterogeneous effects of enterprise nature, life cycle, internal innovation and external support. In addition, this paper examines the moderating effects of total factor productivity and corporate social responsibility, and uses a panel quantile model to test the dynamic effects of core explanatory variables and moderating variables. The marginal contributions of this paper include: (1) shifting the research focus from macro factors affecting common prosperity to micro research on internal prosperity of enterprises, combining macro factors of economic policy uncertainty to explore its internal causal relationship; (2) emphasizing that enterprises actively assume social responsibility and improve total factor productivity will alleviate the negative impact of economic policy uncertainty on internal prosperity of enterprises; (3) using a dynamic perspective to explore the impact of economic policy uncertainty, corporate social responsibility and improving total factor productivity on internal prosperity of enterprises.

The structure of this paper is as follows: In Section 2, we present the relevant literature and theoretical hypotheses. Based on the relevant literature and core issues, we propose three theoretical hypotheses for testing. Section 3 explains the estimation strategy, data sources and variable construction. Section 4 presents the benchmark regression results and a series of robustness tests. Section 5 presents further empirical research results, mainly the results of moderation, heterogeneity and dynamic effects. Section 6 presents conclusions and policy implications, and discusses limitations.

## 2. Theoretical analysis and research hypotheses

### 2.1 The connotation of common prosperity within an enterprise

Iversen and Soskice [7] think common prosperity is not just the result of equal distribution, nor is it the wealth of a few people. It requires the establishment of a coordinated institutional system of primary distribution, redistribution and tertiary distribution, as well as interest relations that are compatible with production relations. Zhi and Shen [8] think this is done to ensure that the fruits of development can be shared by all people and is an effective way to achieve common prosperity. Stakeholder theory is consistent with the goal of common prosperity. The theory holds that stakeholders are individuals, groups or organizations that exchange resources with enterprises and influence each other. As the core of income distribution, enterprises need to consider the needs of multiple stakeholders such as employees, shareholders, government and society, and find a balance between these interests [9, 10]. This article will use stakeholder theory to define the meaning of corporate common prosperity: In the process of income distribution, enterprises should take into account the requirements of all stakeholders and ensure that they can share the fruits of development. In the primary distribution stage, enterprises should protect the interests of employees, customers, shareholders and other partners while pursuing their own interests, such as providing a large number of jobs and high-quality products. In the redistribution stage, enterprises should establish good relations with the government and increase their contributions to taxes in order to provide sufficient funds for redistribution. In the tertiary distribution, enterprises should pay attention to the relationship between the community and the environment, and actively participate in social donations, rural revitalization and public welfare and charity activities.

### 2.2 Economic policy uncertainty and common prosperity within enterprises

Under the current complex international and domestic situation, in order to actively respond to complex changes, ensure high-quality domestic development, and meet the people's yearning for a better life, economic policies have been adjusted frequently, and the predictability has gradually decreased. Overall, the economic policy has significantly increased Uncertainty. Existing research mainly studies the economic consequences of economic policy uncertainty from both macro and micro perspectives.

At the macro level, the impact of economic policy uncertainty on high-quality economic development is obviously negative. It undermines economic resilience, hinders economic growth, and hinders the process of economic recovery by reducing investment, output, and labor efficiency [11]. Scholars also believe that economic policy uncertainty will have an adverse impact on the effective implementation of monetary policy, stock market risks and performance, and high-quality trade development [12]. It is worth noting that the impact of economic policy uncertainty on income disparity is relatively negative. Studies have found that economic policy uncertainty reduces rural investment, has a negative impact on farmers' income growth, and further widens the urban-rural income gap [13].

At the micro level, economic policy uncertainty will have a negative impact on the production and operation of micro entities (enterprises). When operators face the uncertainty of market economic policies, the accuracy of their judgments on future economic development trends will decrease, which means that the expected risk of selecting investment projects will increase, leading to increased difficulty in corporate investment and operation [14]. On the other hand, economic policy uncertainty will also exacerbate the income gap within enterprises. Some scholars believe that unclear economic policies will increase income uncertainty and lead to a widening income gap within enterprises [15]. Al-Thaqeb and Algharabali [16]

find that some scholars have found in their research on executive compensation levels that in periods of greater economic fluctuations and greater economic policy uncertainty, the level of pay fairness is lower.

Combining the above discussion, it is not difficult to find that economic policy uncertainty has a negative impact not only on economic development and fair distribution at the macro level, but also on corporate operating efficiency and income gap at the micro level. For enterprises, achieving common prosperity is no different from the macro level, and requires the elements of "common" and "common prosperity", which are usually referred to as "fair sharing of the pie" and "making the pie bigger".

From an efficiency perspective, the negative impact of economic policy uncertainty on production efficiency is an important factor leading to a decline in the level of common prosperity within an enterprise. Some research has found that China's financial system is mainly controlled by banks. In the financing credit market, banks often impose higher financing requirements on smaller, non-state-owned enterprises [17], which leads to a further reduction in asset allocation efficiency, significantly it affects the investment decisions and credit needs of enterprises, thereby reducing investment efficiency. In terms of labor efficiency, economic policy uncertainty will aggravate the uncertainty faced by enterprises, leading to lower labor allocation efficiency, and low allocation efficiency will lead to lower enterprise productivity levels [18]. In terms of innovation efficiency, Nickell et al. [19] recognized for the first time the important role of R&D expenditure on corporate productivity and attributed it to growth efficiency. Ismail [20] proposed that this growth efficiency occurs because R&D expenditure improves the innovation capabilities of enterprises and has a positive impact on enterprise productivity. Economic policy uncertainty deprives enterprises of a favorable living environment, forcing them to adopt more conservative business strategies. When they reject investment projects, they compete less with external risks and naturally choose to give up or reduce investment in innovation. Enterprises are unwilling to invest too much in innovation and are unable to enjoy the spillover effects and competitive advantages brought by innovation, resulting in reduced production efficiency.

Combined with the above analysis, economic policy uncertainty affects the level of common prosperity of enterprises through multiple channels. First, economic policy uncertainty increases the operating risks of enterprises, causing enterprises to be more conservative in decision-making and reduce investment in innovation and long-term investment. Although this conservative strategy may reduce risks in the short term, in the long run, it limits the development potential and competitiveness of enterprises, thereby affecting the common prosperity of enterprises. Second, economic policy uncertainty increases the financing costs and difficulties of enterprises. Due to policy uncertainty, financial institutions are more cautious in credit assessment of enterprises, resulting in more obstacles for enterprises, especially small and medium-sized enterprises, in the financing process. This not only limits the liquidity of enterprises, but also affects the investment and expansion capabilities of enterprises, thereby affecting the common prosperity of enterprises. Finally, economic policy uncertainty also has a negative impact on the internal income distribution of enterprises. Policy uncertainty increases the operating pressure of enterprises. In order to cope with this pressure, enterprises may take cost-cutting measures, such as reducing employee benefits and salaries. Although this approach can alleviate the financial pressure of enterprises in the short term, in the long run, it will affect the work enthusiasm of employees and the overall production efficiency of enterprises, thereby affecting the common prosperity of enterprises. In summary, this paper proposes research hypothesis 1.

Hypothesis 1: Economic policy uncertainty will reduce the level of common prosperity within enterprises.

## 2.3 Economic policy uncertainty, corporate social responsibility and shared prosperity within enterprises

Corporate Social Responsibility is the behavior of companies allocating resources to stakeholders, expecting to obtain positive returns, and establish direct reciprocal relationships with stakeholders [21]. The establishment of this relationship will significantly reduce the negative impact of uncertain events. From the current research perspective, the potential negative impact of economic policy uncertainty on the level of common prosperity within enterprises can be mitigated. Enterprises that actively assume social responsibility usually achieve risk aversion through "impression management" in a complex and uncertain external economic policy environment [22]. McWilliams et al. [23] and Dai et al. [24] also pointed out that companies that actively assume social responsibility will attract the attention of the government and help them gain advantages in economic policies. This means that the impact of uncertainty on these businesses will be mitigated.

In addition, when the government lacks sufficient funds for public services, enterprises become one of its channels to obtain additional funds [25]. Therefore, in China, companies that actively assume social responsibilities have a rent-seeking function to a certain extent. However, Xiao and Zhang [26] think that this approach is considered safe and effective due to its beneficial external spillover effects. Yuan et al. [27] used data from listed companies from 2008 to 2015 to verify the positive relationship between economic policy uncertainty and the level of corporate social responsibility. They proposed that this relationship is more significant in enterprises and regions with a strong political atmosphere (such as state-owned enterprises, strong political ties, and high political interference).

On the other hand, although the research on the correlation between enterprises' active assumption of social responsibility and common prosperity is still in the stage of practical exploration and is not yet in-depth and systematic, it is not difficult to imagine that as an important part of the socialist market economy, enterprises play an important role in common prosperity by actively assuming social responsibility. Li and Du [28] divided enterprises into three levels according to the degree of corporate social responsibility and linked them to the first, second and third distribution of society. They theoretically explained the profound relationship between social responsibility and social distribution, indicating that corporate social responsibility is a key factor in achieving common prosperity. Enterprises' active assumption of social responsibility can not only promote common prosperity, but also incorporate common prosperity into the mission of corporate social responsibility. This means that enterprises should not only pursue economic benefits, but also incorporate common prosperity into their development goals. Assuming social responsibility can improve internal allocation strategies, improve efficiency, and enhance the operational performance and dynamic capability level of enterprises [29]. Narayan et al. [30] suggested that in order for the social responsibility strategy to be effective, a comprehensive consensus must be reached, that is, from senior management to grassroots employees, they must embrace social responsibility awareness under the goal of common prosperity.

Combined with the above analysis, it is not difficult to think that enterprises that actively assume social responsibility can reduce the negative impact on the level of common prosperity of enterprises in an environment of economic policy uncertainty through various channels. First, enterprises that actively assume social responsibility can enhance their social capital and reputation by establishing a solid reciprocal relationship with stakeholders. This enhanced social capital and reputation can help enterprises obtain more support and resources in the face of economic policy uncertainty, thereby reducing operating risks and maintaining stable development of enterprises. Second, enterprises that actively assume social responsibility can

often attract the attention and support of the government and the public. In an environment of economic policy uncertainty, this attention and support can be transformed into policy preferences and resource tilts, helping enterprises gain competitive advantages in an uncertain environment and reduce the negative impact of uncertainty. Finally, enterprises that actively assume social responsibility can improve employees' work enthusiasm and overall production efficiency by improving internal management and distribution mechanisms. This not only helps enterprises maintain efficient operation in an environment of economic policy uncertainty, but also enhances employees' sense of belonging and satisfaction through fair income distribution and a good working environment, thereby promoting common prosperity within the enterprise. In summary, this paper proposes research hypothesis 2.

Hypothesis 2: Strengthening corporate social responsibility will mitigate the negative impact of economic policy uncertainty on the level of common prosperity within enterprises.

## 2.4 Economic policy uncertainty, total factor productivity and common prosperity within enterprises

In addition to corporate social responsibility, the total factor productivity of a company is also an important factor affecting the level of common prosperity within the company. Total factor productivity refers to the comprehensive efficiency level of various production factors in the production process, reflecting the overall performance of enterprises in technology, management, and innovation [31]. The improvement of total factor productivity means that enterprises can achieve more output with less input, thereby enhancing their profitability and competitiveness.

The improvement of total factor productivity also contributes to the internal income distribution of enterprises. Enterprises can motivate and retain outstanding talents by increasing employee salaries, providing more benefits and rewards, improving the working environment, and promoting career development, thereby achieving common prosperity within the enterprise [28]. In addition, the improvement of total factor productivity can alleviate the negative impact of economic policy uncertainty on the level of common prosperity within enterprises, as it enhances their ability to resist risks and adapt. This enables enterprises to maintain stable production and operation in uncertain environments, reducing investment shocks, production interruptions, and decreased competitiveness caused by policy changes [32].

The improvement of total factor productivity can alleviate the negative impact of economic policy uncertainty on the level of common prosperity within enterprises through various means. Firstly, the improvement of total factor productivity means an overall increase in the efficiency of enterprises in technology, management, and innovation, which enables them to achieve higher output and benefits in resource limited situations, thereby enhancing their profitability and market competitiveness. In an environment of uncertain economic policies, this enhanced profitability and competitiveness help businesses resist external risks and maintain stable development. Secondly, the improvement of total factor productivity can enhance the internal income distribution mechanism of enterprises. By improving production efficiency, enterprises can gain more profits, thus having the ability to increase employee wages and benefits, improve the working environment, and promote employee career development. This not only helps motivate and retain outstanding talents, but also improves employees' work enthusiasm and satisfaction, thereby promoting common prosperity within the enterprise. Finally, the improvement of total factor productivity can enhance the adaptability and flexibility of enterprises. In an environment of uncertain economic policies, enterprises need

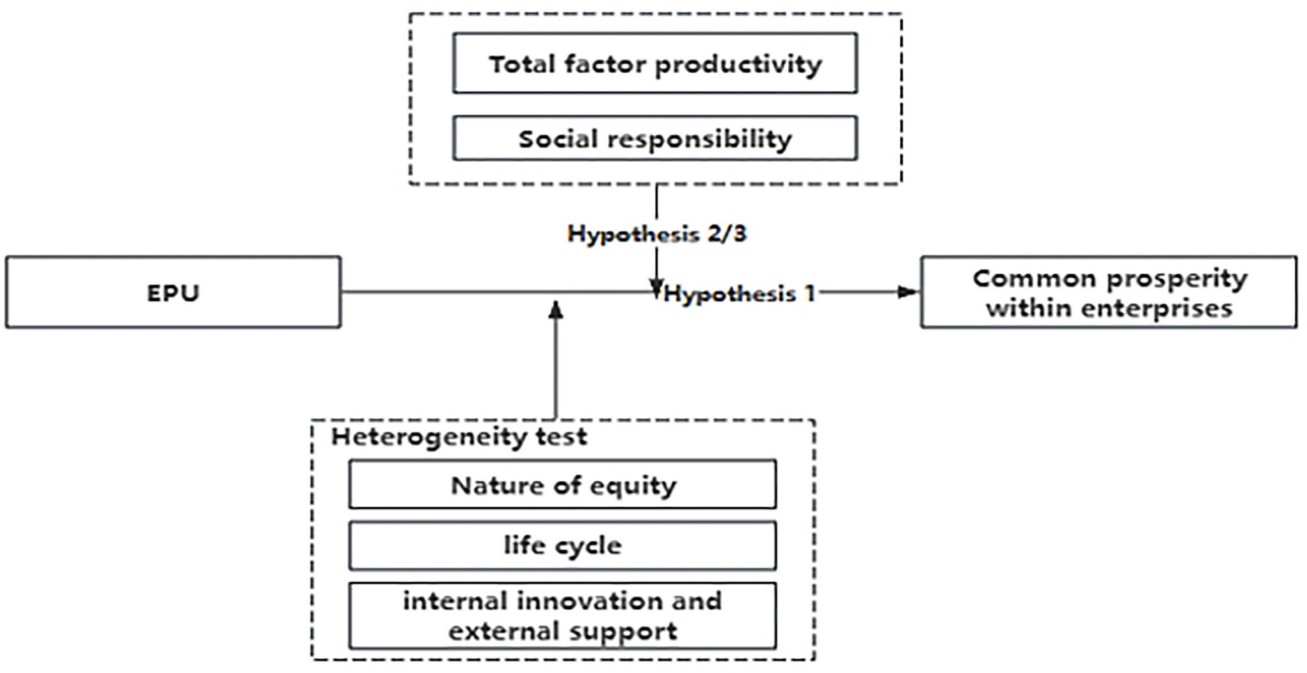

**Fig 1. Conceptual framework.**

to have the ability to quickly adjust and respond to changes. The improvement of total factor productivity enables enterprises to adjust their production and operation strategies more flexibly in the face of policy changes, reduce the negative impact caused by policy changes, and thus maintain the stable development of enterprises. In summary, this article proposes research hypothesis 3.

Hypothesis 3: The improvement of total factor productivity will mitigate the negative impact of economic policy uncertainty on the level of common prosperity within enterprises.

Fig 1 is the conceptual framework diagram described in this article, in which Hypothesis 1 represents the inhibitory effect of EPU on common prosperity within the enterprise, Hypothesis 2 represents that social responsibility can weaken the inhibitory effect of EPU on common prosperity within the enterprise, and Hypothesis 3 represents that total factor productivity can weaken the inhibitory effect of EPU on common prosperity within the enterprise.

## 3. Study design

### 3.1 Sample selection and data source

This paper focuses on companies listed on the Shanghai and Shenzhen A-share markets, covering the time period from 2011 to 2020. Initial data underwent the following operations: (1) Excluding samples from the financial industry;(2) Removing companies in ST or PT status;(3) Excluding samples with less than 5 years of trading history and companies with abnormal financial data (negative income, insolvency). The primary data sources for this paper are the CSMAR and WIND databases. To minimize the impact of extreme outliers on empirical results, all continuous variables were Winsorized at the 1% and 99% levels. The final dataset comprises 22,321 unbalanced panel observations.

### 3.2 Variable definitions

**3.2.1 Dependent variable: Enterprise common prosperity (lncps).** The dependent variable of this paper is the level of common prosperity of enterprises. This paper uses CSMAR and the Sustainable Development and Corporate Social Responsibility Research Team of East China Normal University to measure the level of common prosperity of enterprises. Jian [33] found that the research constructed a theory and framework for enterprises to participate in achieving common prosperity. In the primary distribution, enterprises bear social responsibilities such as providing jobs, reasonable remuneration, and rights protection for employees; in the redistribution, enterprises bear social responsibilities such as providing taxes to the government; at the same time, enterprises play an important role in the tertiary distribution by participating in social welfare, poverty alleviation, and charitable donations. Therefore, the enterprise common prosperity score used in this paper is measured based on multiple dimensions. For details, please refer to Table 1. The indicator system includes three first-level indicators such as primary distribution, redistribution, and tertiary distribution, 9 second-level indicators, and 37 third-level indicators, which can fully reflect the level of enterprise

**Table 1. Enterprise common prosperity scoring index system.**

| Primary Indicator | Secondary Indicator | Tertiary Indicator |
|---|---|---|
| Initial Distribution | Employee Employment | Year-end Number of Employees<br>New Job Positions<br>Gender Diversity in Management<br>Job Competitiveness and Career Management<br>Care for Vulnerable Groups |
| | Employee Compensation | Per Share Salary Contribution<br>Employee Profit Sharing<br>Average Salary<br>Employee Salary Growth Rate<br>Average Salary Ratio between Directors/Supervisors and Other Employees |
| | Employee Welfare | Legal Employment<br>Safety Production Investment<br>Safety Production Level<br>Occupational Health Protection<br>Employee Social Security Fund Contribution Ratio<br>Commercial Insurance<br>Employee Rights Protection |
| | Customer Sharing | Product/Service Quality<br>Product Recall Quantity<br>Consumer Rights Protection |
| | Shareholder Sharing | Return on Net Assets<br>Per Share Cash Dividend<br>Minority Shareholder Rights Protection (Independent Director System) |
| | Partner Sharing | Distributor Rights Protection<br>Per Share Interest Contribution<br>Debtor Rights Protection<br>Supplier Rights Protection<br>Community Ecological Environment Protection |
| | Fair Competition | Competitor Cooperation/Alliance<br>Fair Competition/Anti-Unfair Competition |
| Redistribution | Tax Contribution | Per Share Tax Contribution<br>Average Income Tax<br>Legal Tax Payment |
| Tertiary Distribution | Charitable Contribution | Public Welfare and Charity<br>Poverty Alleviation Investment<br>Number of Sponsored Poor Students<br>Poverty Alleviation and Rural Revitalization Investment |

participation in achieving common prosperity. Among them, each sub-indicator is weighted after reverse indicator adjustment, missing value filling, standardization, and indicator mapping to obtain the enterprise common prosperity score. The higher the score, the higher the level of enterprise common prosperity. After scoring each aspect, enterprises are divided into 9 grades (CCC is the lowest and AAA is the highest) according to the comprehensive score, which is quantified from 1 to 9 and serves as an alternative indicator of common prosperity within enterprises in this article.

**3.2.2 Independent variable: Economic policy uncertainty.** This paper utilizes the monthly Economic Policy Uncertainty index published by Baker et al. [34] The index is based on news reporting data from the Hong Kong-based South China Morning Post. After filtering the data, a text analysis approach is employed to construct the Economic Policy Uncertainty index for China. To better match the data, the annual arithmetic mean of the index is calculated, converting monthly economic policy uncertainty into annual economic policy uncertainty.

**3.2.3 Control variables.** • Growth: Represented by the growth rate of operating income.

- ROE (Return on Equity): Represented by the return on equity.

- Loss: Based on the net profit of the company for the year; 1 if greater than 0, otherwise 0.

- Top5 (Top 5): Represents the shareholding ratio of the top five major shareholders.

- TobinQ: Measured as the ratio of the company's market value to replacement cost.

- Mfee (Management Fee): Determined by the ratio of management fees to main operating income.

- Balance: Represents the ratio of shareholding by the second to fifth largest shareholders to the first largest shareholder. Table 2 provides descriptive statistics for the variables.

## 3.3 Model design

To effectively examine the relationship between economic policy uncertainty and common prosperity within enterprises and alleviate potential endogeneity issues, this paper constructs the following two-way fixed-effects model.

$$lncps_{i,t} = \beta_0 + \beta_1 lnEPU_{i,t} + \beta_i CV_{i,t} + \sum Year + \sum id + \varepsilon_{i,t} \qquad (1)$$

Here, the subscript $i$ denotes the company, t represents the year, and $\sum$ Year and $\sum$ id represent controls simultaneously accounting for both time and individual effects. *lncps* signifies the

**Table 2. Descriptive statistics of variables.**

| Variable symbol | Sample size | Average | Standard deviation | Minimum value | Maximum value |
|---|---|---|---|---|---|
| lnEPU | 22321 | 4.945 | 0.0860 | 4.829 | 5.110 |
| lncps | 22321 | 1.433 | 0.645 | 0 | 2.197 |
| ROE | 22321 | 0.057 | 0.138 | -1.072 | 0.397 |
| Cashflow | 22321 | 0.046 | 0.068 | -0.200 | 0.257 |
| Loss | 22321 | 0.112 | 0.316 | 0 | 1 |
| Top5 | 22321 | 0.523 | 0.151 | 0.175 | 0.892 |
| Balance | 22321 | 0.352 | 0.285 | 0.00600 | 1 |
| TobinQ | 22321 | 2.065 | 1.444 | 0.802 | 17.73 |
| Mfee | 22321 | 0.093 | 0.080 | 0.0080 | 0.766 |

level of common prosperity within the enterprise, and *lnEPU* represents the level of economic policy uncertainty. If the coefficient on $\beta_1$ is significantly negative, it indicates that the increase in economic policy uncertainty hinders the enterprise's common prosperity. Finally, to mitigate endogeneity concerns on the experimental results, logarithmic transformations were applied to both the main explanatory variables and the dependent variable.

# 4. Empirical testing and results analysis

## 4.1 Basic regression analysis

Table 3 presents the baseline regression results on the impact of economic policy uncertainty on the level of common prosperity within enterprises. In column (1), it is evident that without considering control variables, the coefficient of economic policy uncertainty on the internal common prosperity of enterprises is -1.627, significantly negative at the 1% level. This indicates that without controlling for other factors, an increase in economic policy uncertainty will significantly decrease the level of common prosperity within the enterprise.

In column (2), considering control variables, it is apparent that the coefficient of economic policy uncertainty on the internal common prosperity of enterprises remains significantly negative. This implies that Hypothesis 1, stating that "economic policy uncertainty will reduce the level of common prosperity within the enterprise," is supported by the empirical evidence.

**Table 3. The impact of economic policy uncertainty on internal common prosperity within enterprises.**

|  | (1) | (2) |
|---|---|---|
| lnEPU | -1.627*** | -3.168*** |
|  | (0.501) | (0.509) |
| ROE |  | 0.383*** |
|  |  | (0.032) |
| Cashflow |  | 0.367*** |
|  |  | (0.051) |
| Loss |  | 0.023* |
|  |  | (0.013) |
| Top5 |  | 0.534*** |
|  |  | (0.043) |
| Balance |  | -0.048** |
|  |  | (0.020) |
| TobinQ |  | -0.018*** |
|  |  | (0.003) |
| Mfee |  | -0.134** |
|  |  | (0.063) |
| Intercept term | 9.364*** | 16.710*** |
|  | (2.466) | (2.507) |
| Individual effect | Yes | Yes |
| Time effect | Yes | Yes |
| Adj-R2 | 0.042 | 0.070 |
| N | 22321 | 22321 |

Note: *, **, *** represent significance at the 10%, 5% and 1% levels, and the standard errors in parentheses are the same below.

**Table 4. Robustness test: Controlling for exogenous shocks.**

|  | (1) | (2) |
|---|---|---|
|  | **Excluding the impact of China's stock market crash** | **Remove stock market crash and municipality samples** |
| lnEPU | -3.227*** | -3.003*** |
|  | (0.517) | (0.560) |
| Intercept term | 17.019*** | 15.884*** |
|  | (2.544) | (2.755) |
| Control variables | Yes | Yes |
| Individual effect | Yes | Yes |
| Time effect | Yes | Yes |
| Adj-R2 | 0.061 | 0.063 |
| N | 19816 | 17408 |

## 4.2 Robustness tests

**4.2.1 Controlling for exogenous shocks.**   In the current economic development process, the performance of the financial market undoubtedly profoundly influences the progress of common prosperity in China. Significant financial shocks can lead to the overall loss and redistribution of wealth for the nation, investors, and even economic entities. Neglecting such factors may result in endogeneity issues in the research findings. During the selected time interval in this study, there was a significant financial shock, namely the Chinese stock market crash in 2015. It is challenging to account for such factors through variable construction. In view of this, this paper deletes the sample of enterprises in 2015 to eliminate the impact of China 's stock market crash. On the other hand, considering that municipalities have great economic and political particularities, there may be great differences between supply chain financing capacity and high-quality development level of enterprises. In this regard, this paper excludes the sample of municipalities directly under the central government. The results of Table 4 show that the core conclusion of this paper, "economic policy uncertainty reduces the level of common prosperity within enterprises," has not changed.

## 4.3 One-lag and bootstrap method

In order to further test the robustness of the core conclusions, this paper uses the explanatory variable (economic policy uncertainty) to lag one period to test the correlation between economic policy uncertainty and common prosperity within the enterprise. In order to weaken the interference of sample self-selection on the empirical results, this paper further uses the Bootstrap method to randomly sample 500 times, and estimates the parameters of the samples after sampling. The specific regression results are shown in Table 5. It can be seen from Table 5 that the significance and direction of the explanatory variables have not changed significantly, and the research conclusions are still robust.

## 4.4 Instrumental variable method

From the perspective of micro-enterprises, economic policy uncertainty is not directly generated, so it is relatively exogenous. However, as a basic component of China's macro-economy, the overall performance of enterprises often determines the direction of macroeconomic policy adjustments. As the core representative of micro-enterprises, listed companies reflect the development of China's economy to a certain extent, so there may be reverse causality. In

**Table 5. Robustness test: Lag one period and bootstrap.**

|  | (1) | (2) | (3) | (4) |
|---|---|---|---|---|
|  | lag one period | lag one period | bootstrap | bootstrap |
| lnEPU(-1) | -0.112 | -0.500*** |  |  |
|  | (0.151) | (0.153) |  |  |
| lnEPU |  |  | -1.627*** | -3.168*** |
|  |  |  | (0.576) | (0.675) |
| Intercept term | 1.918*** | 3.522*** | 9.364*** | 16.710*** |
|  | (0.738) | (0.744) | (2.837) | (3.318) |
| Control variables | No | Yes | No | Yes |
| Individual effect | Yes | Yes | Yes | Yes |
| Time effect | Yes | Yes | Yes | Yes |
| Adj-R2 | 0.040 | 0.068 | 0.042 | 0.070 |
| N | 18812 | 18362 | 22321 | 22321 |

order to deal with this endogeneity problem, this paper draws on the method of Dong and Liu [35] and uses US economic policy uncertainty (Lnepum_IV) as an instrumental variable for two-stage least squares (2SLS) estimation.

The selection of instrumental variables usually needs to consider their correlation and exogenous conditions. This paper selects US economic policy uncertainty as an instrumental variable for China's economic policy uncertainty based on these two conditions: First, with the continuous expansion of the global value chain, external shocks have gradually become an important source of macroeconomic fluctuations. Changes in US economic policies can quickly affect China's economic environment, thereby affecting the formulation and implementation of China's economic policies. There is a correlation between the two. Second, it is difficult for US economic policies to directly affect the production and operation choices of Chinese enterprises across the Chinese government and the Chinese economic environment, and it is also impossible to directly affect the level of common prosperity within enterprises. For the US EPU to play a role, it can only affect the level of common prosperity within enterprises through the uncertainty of China's economic policies, that is, there is exogeneity between the two.

In addition, the US economic policy uncertainty index used in this paper is derived from the index constructed by Baker et al. [33]. This index captures information containing economy, policy and uncertainty based on media information through text analysis and standardizes it to obtain the economic policy uncertainty index. The index mainly consists of three parts: the first part is obtained through standardized economic and policy uncertainty-related vocabulary from 10 major newspapers in the United States; the second part considers temporary taxes as a source of uncertainty for enterprises and households, drawing on the report of the Congressional Budget Office (CBO); the third part considers the impact of monetary and fiscal policies, drawing on the professional forecast survey of the Federal Reserve Bank of Philadelphia. The US economic policy uncertainty index is based on monthly data. This paper uses the geometric mean of monthly data as the annual data as an instrumental variable.

Table 6 reports the IV estimation results of the two-stage least squares method. Among them, (1) is listed as the first-stage regression. The results show that US economic policy uncertainty has a significant positive impact on China's economic policy uncertainty (0.0977), which meets the correlation requirements of the instrumental variable; column (2) is the second-stage regression result. EPU has a significant negative impact on the common prosperity within the enterprise, which is consistent with the conclusion of the previous article. The

**Table 6. Robustness test: Instrumental variable test.**

| | (1) | (2) |
|---|---|---|
| | **The first stage** | **The second stage** |
| lnEPU | | -3.903*** |
| | | (0. 243) |
| Lnepum_IV | 0.0977*** | |
| | (0.003) | |
| Control variables | Yes | Yes |
| Individual effect | Yes | Yes |
| Time effect | Yes | Yes |
| LM statistic (p-value) | 0.00 | |
| Kleibergen-Paap rk Wald F | 256.79 | |
| N | 18362 | 18362 |

Kleibergen-Paap rk Wald F value is 256.79, and the LM statistic is significant at the 1% level, indicating that the risk of weak instrumental variables is very small, and we have selected appropriate instrumental variables. The results in Table 6 show that after using the instrumental variable method to deal with the endogeneity problem of the model, our research results are still valid.

## 5. Further research

### 5.1 Moderation effect test

Based on the theoretical analysis in the previous article, this paper further examines the moderating effect of economic policy uncertainty on the internal prosperity of enterprises, and uses the following adjustment variables for regression: Total factor productivity (lnlp) measures the level of enterprise production efficiency and is constructed using the LP method. The construction process is as follows: (1) select the sales revenue, net fixed assets, employee wages, and intermediate inputs in the production process of manufacturing enterprises as explanatory variables; (2) perform fixed effect regression after fixing individual effects and annual effects, and obtain the residual term at the same time; (3) take the logarithm of the residual term to obtain the total factor productivity proxy indicator. Corporate social responsibility (lnsr) measures the fulfillment of corporate social responsibility. This paper uses the professional evaluation system of listed company social responsibility reports constructed by "Hexun.com" to represent it. In order to avoid endogeneity problems, this paper takes the logarithm of corporate social responsibility.

Table 7 reports the analysis results of the moderating effect of total factor productivity and social responsibility. Column (1) is the estimated result of adding the moderating variable total factor production effect to the baseline regression results. Column (2) adds the interaction term between the moderating variable total factor productivity and economic policy uncertainty to column (1). The regression coefficient of the interaction term is -0.510, which is significant at the 5% significance level. Column (3) is the estimated result after adding the moderating variable social responsibility to the baseline regression. Column (4) adds the interaction term between social responsibility and economic policy uncertainty to column (3). The regression coefficient of the interaction term is -0.172, which is significant at the 5% significance level. The above results show that total factor productivity growth and corporate social responsibility will weaken the negative effect of economic policy uncertainty on common prosperity, thus supporting Hypothesis 2 and Hypothesis 3 of this paper. The reason is that actively

**Table 7. Adjustment test results.**

|  | (1) | (2) | (3) | (4) |
|---|---|---|---|---|
|  | Total factor productivity | | Social responsibility | |
| lnlp | 2.655*** | 5.181*** |  |  |
|  | (0.073) | (1.227) |  |  |
| EPU_lp |  | -0.510** |  |  |
|  |  | (0.247) |  |  |
| lnsr |  |  | 0.166*** | 1.022** |
|  |  |  | (0.012) | (0.428) |
| EPU_sr |  |  |  | -0.172** |
|  |  |  |  | (0.086) |
| lnEPU | 2.486*** | 3.663*** | -4.383*** | -3.801*** |
|  | (0.516) | (0.770) | (0.515) | (0.592) |
| Intercept term | -17.078*** | -22.911*** | 22.097*** | 19.197*** |
|  | (2.596) | (3.839) | (2.527) | (2.913) |
| Individual effect | Yes | Yes | Yes | Yes |
| Time effect | Yes | Yes | Yes | Yes |
| Adj-R2 | 0.131 | 0.132 | 0.079 | 0.079 |
| N | 21792 | 21792 | 21789 | 21789 |

assuming social responsibility can not only help enterprises establish a sense of responsibility and sharing awareness and enhance social credibility, but also enhance the adaptability of enterprises, so that enterprises can maintain stable development in the face of an uncertain external environment. In addition, improving the total factor productivity of enterprises is a reflection of their ability to continuously innovate technology, which helps enterprises cope with market competition in an uncertain environment, expand economic scale, and achieve common prosperity.

The significance of this research result is not limited to the specific context of Chinese listed companies. Economic policy uncertainty is a global phenomenon that affects companies in various industries and countries. For example, research shows that economic policy uncertainty significantly inhibits foreign direct investment in Asian countries, thereby having a negative impact on the production, operation and financing of companies in the country [36]. Similarly, the impact of economic policy uncertainty on trade and sustainable economic development is reflected in different industries [37]. In industries such as manufacturing and high-tech industries, improving total factor productivity through continuous technological innovation can help companies better cope with an uncertain economic environment. This is because higher productivity levels enable companies to maintain their competitive advantage and adapt to market changes more effectively. In addition, corporate social responsibility plays a vital role in enhancing organizational resilience in different industries. Research shows that companies with strong corporate social responsibility practices are better able to cope with external shocks, such as the COVID-19 epidemic. corporate social responsibility initiatives help build trust and credibility with stakeholders, thereby enhancing the ability of companies to recover from crises and maintain stable operations.

## 5.2 Heterogeneity test

**5.2.1 Equity nature perspective.** Considering that state-owned enterprises and non-state-owned enterprises may face different national economic policies, this study further divides the sample into state-owned enterprises and non-state-owned enterprises based on the nature of

**Table 8. Heterogeneity test: Nature of equity.**

|  | (1) | (2) |
|---|---|---|
|  | **State-owned enterprises** | **Non-state-owned enterprises** |
| lnEPU | -3.428*** | -2.824*** |
|  | (0.676) | (0.796) |
| Intercept term | 18.051*** | 14.986*** |
|  | (3.332) | (3.913) |
| Control variables | Yes | Yes |
| Individual effect | Yes | Yes |
| Time effect | Yes | Yes |
| Adj-R2 | 0.060 | 0.073 |
| N | 8120 | 13693 |
| Empirical P value | 0.000 |  |

Note: The empirical P value is used to test the significance of differences between groups and is calculated using Fisher's combined test (sampling 1000 times), the same below.

equity. State-owned enterprises refer to enterprises whose major shareholders (actual controllers) belong to the central or local SASAC, state organs, state-owned enterprises and institutions, and the rest are non-state-owned enterprises. The results are shown in Table 8. According to the estimates of columns (1) and (2), the coefficients of the state-owned enterprise and non-state-owned enterprise groups are significantly negative. However, the coefficient of the state-owned enterprise group is significantly larger than that of the non-state-owned enterprise group, and the empirical p value is significantly lower than the 1% level. This means that for state-owned enterprises, changes in national economic policies directly affect their decision-making and corporate performance, thereby affecting the level of common prosperity within the enterprise. Non-state-owned enterprises are relatively less affected by such changes. This conclusion also has certain reference value in other countries and industries. The impact of economic policy uncertainty on corporate prosperity may show similar patterns in different countries and industries. Enterprises with different organizational structures may show different responses when dealing with economic policy uncertainty. State-owned enterprises may have greater advantages in dealing with policy changes due to their special resources and support, while non-state-owned enterprises may face more challenges and risks [38]. Therefore, the results of this study have a certain degree of generalizability across countries and industries, but the specific circumstances and differences between countries and industries need to be taken into account.

**5.2.2 Potential limitations and biases.** There are some potential limitations and biases when comparing state-owned enterprises and non-state-owned enterprises as different groups in the study. First, state-owned enterprises usually enjoy government support and resources, including preferential loans, fiscal subsidies, and policy preferences, which enable state-owned enterprises to show greater flexibility and adaptability in dealing with economic policy uncertainties. In contrast, non-state-owned enterprises may face more challenges and restrictions in resource acquisition and market competition. Due to the lack of direct government support, non-state-owned enterprises may show higher risks and uncertainties in dealing with economic policy uncertainties. In addition, there are inherent differences in governance structure and goals between state-owned enterprises and non-state-owned enterprises. State-owned enterprises often face political interference, and the policy goals they need to achieve may not be consistent with maximizing shareholder wealth, which may lead to inefficiency and

different responses to economic policies. Non-state-owned enterprises usually aim to maximize shareholder wealth, have relatively simple governance structures, and may have more efficient decision-making processes. Finally, mixed-ownership enterprises have the characteristics of both state-owned and non-state-owned enterprises, which may be ignored in the study. The existence of mixed-ownership enterprises makes it possible that simply dividing enterprises into state-owned and non-state-owned enterprises may not cover all types of enterprises. These limitations and biases must be taken into account in research to ensure a more accurate assessment of the impact of economic policy uncertainty on firm-internal prosperity.

**5.2.3 Life cycle perspective.**   Differences in the life cycle of an enterprise often affect its business decisions and corporate benefits, and the economic policies implemented by the government for enterprises in different life cycles will also be different, and this difference often greatly affects the development of an enterprise. In other words, the uncertainty of the country's economic policies for related enterprises is likely to have different impacts on enterprises in different life cycles. In view of this, this paper draws on the research of Liang et al. [36] and uses variables such as sales revenue growth rate, retained earnings rate, capital expenditure rate, and company age to score after sorting by industry, and calculates the company's comprehensive score on the four indicators. The comprehensive score is further grouped for regression, with the top 25% being growth companies, the bottom 25% being decline companies, and the rest being mature companies.

The results are shown in Table 9. Throughout the life cycle, economic policy uncertainty has a significant negative impact on the common prosperity within the enterprise, and the empirical P value is significant at the 1% level. As enterprises transition from the growth stage to the decline stage, the negative impact of economic policy uncertainty on enterprises becomes smaller and smaller, highlighting the importance of economic policies in achieving the common prosperity of enterprises. The government often invests a lot of policy support in growth-stage enterprises, but as enterprises transition to recession, policy support continues to decrease, and the impact of economic policies on enterprises becomes uncertain and gradually weakens. This result suggests that the government should provide clear economic policy support for growth-stage or mature-stage enterprises to encourage them to achieve the goal of common prosperity for enterprises.

**5.2.4 Internal innovation and external support perspective.**   The enhancement of internal innovation capabilities within a company can stimulate labor efficiency, improve income levels, and enhance income distribution, thereby achieving common prosperity within the enterprise [39]. Additionally, from an external perspective, government subsidies to businesses

**Table 9. Heterogeneity test: Life cycle.**

|  | (1) | (2) | (3) |
|---|---|---|---|
|  | Growth stage | Maturity stage | Decline stage |
| lnEPU | -9.152*** | -3.478*** | -2.416*** |
|  | (1.907) | (0.793) | (0.869) |
| Intercept term | 46.381*** | 18.279*** | 12.955*** |
|  | (9.372) | (3.904) | (4.282) |
| Control variables | Yes | Yes | Yes |
| Individual effect | Yes | Yes | Yes |
| Time effect | Yes | Yes | Yes |
| Adj-R2 | 0.061 | 0.065 | 0.066 |
| N | 3216 | 10464 | 8136 |
| Empirical P value | (1) (2) is 0.000; (2) (3) is 0.005; (1) (3) is 0.000 | | |

**Table 10. Heterogeneity test: Internal innovation and external support.**

| | (1) | (2) | (3) | (4) |
|---|---|---|---|---|
| | Internal innovation | | External support | |
| | High | Low | High | Low |
| lnEPU | -2.706*** | -3.329*** | -2.389*** | -3.281*** |
| | (0.838) | (0.701) | (0.630) | (1.065) |
| Intercept term | 14.450*** | 17.513*** | 13.003*** | 17.121*** |
| | (4.115) | (3.452) | (3.097) | (5.248) |
| Control variables | Yes | Yes | Yes | Yes |
| Individual effect | Yes | Yes | Yes | Yes |
| Time effect | Yes | Yes | Yes | Yes |
| Adj-R2 | 0.086 | 0.056 | 0.057 | 0.075 |
| N | 10918 | 10874 | 13813 | 7979 |
| Empirical P value | 0.000 | | 0.000 | |

significantly improve their operational conditions, enhance profitability, increase the share of labor income, and lay the foundation for achieving strategic goals of common prosperity through the formation of a rational income distribution pattern [40].

To further examine the roles of internal innovation capabilities and the magnitude of external government subsidy levels in the impact of economic policy uncertainty on common prosperity within enterprises, this paper uses the ratio of research and development (R&D) expenditure to operating income and the ratio of total government subsidies to operating income as proxies for internal innovation and external support, respectively. The samples are then divided into high and low groups based on the median values of each indicator, and group regression analyses are conducted. The results, as shown in Table 10, reveal that in the context of internal innovation, for both the high and low groups, the coefficient of economic policy uncertainty (lnEPU) is significantly negative at the 1% level, with empirical p-values being highly significant. However, compared to the group with higher internal innovation capabilities, the negative impact of economic policy uncertainty on common prosperity within enterprises is more pronounced in the group with lower internal innovation capabilities. Concerning external support, the results indicate that, similar to internal innovation, both high and low groups exhibit significantly negative coefficients for economic policy uncertainty (lnEPU) at the 1% level, with highly significant empirical p-values. Here too, the negative impact of economic policy uncertainty on common prosperity within enterprises is more pronounced in the group with lower external support levels. These findings suggest that the strengthening of both external support and internal innovation will be beneficial for enterprises to mitigate the adverse effects of economic policy uncertainty on common prosperity.

## 5.3 Dynamic feature examination

In the previous sections, we examined the moderating effects of total factor productivity and corporate social responsibility on the relationship between economic policy uncertainty and common prosperity within enterprises. We further recognize that the positive effects of total factor productivity and corporate social responsibility on common prosperity may not be a static process. The positive impact of total factor productivity or the negative impact of economic policy uncertainty may dynamically change with fluctuations in the current level of common prosperity within the enterprise. To better investigate this process, we conduct panel quantile estimations at percentiles 10%, 25%, 50%, 75%, and 90%.

**Table 11. Dynamic feature inspection.**

| | (1) | (2) | (3) | (4) | (5) |
|---|---|---|---|---|---|
| | **10%** | **25%** | **50%** | **75%** | **90%** |
| lnlp | 3.716*** | 3.127*** | 2.263*** | 1.799*** | 1.349*** |
| | (0.100) | (0.075) | (0.045) | (0.032) | (0.031) |
| lnsr | 0.124*** | 0.196*** | 0.270*** | 0.383*** | 0.430*** |
| | (0.011) | (0.011) | (0.016) | (0.026) | (0.035) |
| lnEPU | -4.665*** | -5.043*** | -2.590*** | -0.896* | -0.016 |
| | (1.719) | (1.298) | (0.775) | (0.544) | (0.534) |
| Intercept term | 13.133 | 17.068*** | 7.895** | 1.200 | -1.613 |
| | (8.468) | (6.396) | (3.817) | (2.681) | (2.629) |
| Control variables | Yes | Yes | Yes | Yes | Yes |
| Individual effect | Yes | Yes | Yes | Yes | Yes |
| Time effect | Yes | Yes | Yes | Yes | Yes |
| R2 | 0.2368 | 0.2148 | 0.1725 | 0.1805 | 0.1401 |
| N | 21789 | 21789 | 21789 | 21789 | 21789 |

As shown in Table 11, firstly, we focus on the coefficient of economic policy uncertainty (lnEPU) on common prosperity within enterprises (lncps). The coefficient is significantly negative at the 10% level for all percentiles except the 90th percentile. Moreover, as the percentile increases, the significance and the absolute value of the coefficient decrease. This suggests that, as the level of common prosperity within enterprises rises, the negative impact of economic policy uncertainty gradually diminishes, becoming statistically insignificant at the 90th percentile. These results indicate the dynamic features of economic policy uncertainty (lnEPU) on common prosperity within enterprises.

Second, we examine the impact of corporate social responsibility (lnsr) on intra-firm shared prosperity (lncps). This variable is significantly positive at the 1% level for all percentiles, and the coefficient increases with increasing percentiles. This means that as the level of common prosperity within an enterprise increases, the positive impact of corporate social responsibility gradually increases, forming a positive interaction. These findings also support the dynamic nature of Corporate Social Responsibility.

Finally, we examine the impact of total factor productivity (lnlp) on intra-firm shared prosperity (lncps). In contrast to Corporate Social Responsibility, this variable is significantly positive at the 1% level for all percentiles, but the coefficient decreases as the percentile increases. This shows that as the level of common prosperity within the enterprise increases, the positive impact of total factor productivity gradually weakens, but the influence has always remained at a high level, highlighting the importance of total factor productivity to the sustainable development of the enterprise and achieving common prosperity within the enterprise. These findings also support the dynamic characteristics of total factor productivity.

Combining the empirical results, we observe that as the level of common prosperity of enterprises continues to increase, the impact of total factor productivity and economic policy uncertainty gradually weakens, and the positive impact of corporate social responsibility gradually increases. Horizontally, total factor productivity (prosperity factor) maintains a higher influence than corporate social responsibility (common factor) and economic policy uncertainty. This means that under the current economic situation, the effectiveness of enterprises in enlarging the "cake" is still greater than the effectiveness of equitably distributing the "cake". Only when corporate profitability and production efficiency are significantly improved can relevant stakeholders gain maximum benefits. On the other hand, with the improvement of

the level of common prosperity, the negative impact of economic policy uncertainty has been significantly reduced, and the role of enterprises in actively assuming social responsibility has become more and more prominent in their sustainable development, highlighting the importance of paying attention to corporate social responsibility in an uncertain environment significance.

# 6. Conclusion

## 6.1 Conclusion and recommendations

As China enters a new era, it is an inevitable requirement to achieve common prosperity. In the face of complex domestic and foreign forms, economic recovery is imperative, frequent changes in economic policies, which are important for the realization of common prosperity of the main enterprises, facing many uncertainties. Based on this, this paper studies its impact on the common prosperity of enterprises from the perspective of economic policy uncertainty, and examines various negative and positive factors under this influence. Specifically, starting from the perspective of economic policy uncertainty, this paper first analyzes the negative impact of economic policy uncertainty on common prosperity within enterprises at the theoretical level. It conducts a mechanism-level analysis from the angles of total factor productivity and corporate social responsibility. Subsequently, using the non-financial listed companies on the Shanghai and Shenzhen stock exchanges from 2011 to 2020 as samples, the paper empirically tests the impact of economic policy uncertainty on common prosperity within enterprises. It conducts multidimensional analyses focusing on aspects like equity nature, life cycle, internal innovation, and external support. Furthermore, it explores the heterogeneity features of economic policy uncertainty on common prosperity within enterprises from a dynamic perspective.

The research results show that economic policy uncertainty has a significant negative impact on internal corporate prosperity. Although other exogenous shocks are controlled, the model settings are changed, the parameter estimation method is changed, and the endogeneity problem is solved, the conclusion that economic policy uncertainty significantly reduces the level of mutual prosperity within enterprises still holds. From the perspective of heterogeneity analysis, the negative impact of economic policy uncertainty on common prosperity is more obvious among state-owned enterprises, enterprises in the growth life cycle, enterprises with poor innovation capabilities and low external support. From the perspective of moderating effects, enhancing corporate social responsibility and total factor productivity can help alleviate the above-mentioned negative impacts. From the perspective of dynamic characteristics, as the level of common prosperity within an enterprise increases, the negative impact of economic policy uncertainty and the positive impact of total factor productivity gradually weaken, while the positive impact of corporate social responsibility gradually increases.

Based on the above research conclusions, the following policy recommendations are proposed:

Enhancing Administrative Capacities and Strategic Planning: Given the negative moderating effect of total factor productivity on economic policy uncertainty, especially for companies receiving significant government subsidies and with strong innovation capabilities, it is essential for the government and enterprises to jointly address this issue. The government should focus on improving its administrative capabilities, development planning, and social credibility under complex social backgrounds. Prudent decision-making and consistency in policy implementation are crucial. Using effective economic and policy measures, the government should improve the business environment, support "sunrise" industries, and encourage companies with promising development trends, fostering innovation and enhancing production efficiency. Ultimately, this will help companies grow and achieve common prosperity goals.

Encouraging Corporate Social Responsibility: Since corporate social responsibility plays a negative moderating role in reducing economic policy uncertainty's impact on common prosperity, it effectively alleviates the negative effects. Therefore, the government should encourage, support, and guide companies to invest in social responsibility. Establishing sound platforms and policies for corporate social responsibility is vital. For companies, recognizing that taking on social responsibility is not a burden but a way to effectively mitigate the negative impact of "economic policy uncertainty" is crucial. Actively practicing corporate social responsibility strengthens the awareness of common prosperity among employees and other stakeholders, fostering closer ties within and outside the company.

Addressing Constraints on State-Owned Enterprises (SOEs) and Supporting Growth Enterprises: As economic policy uncertainty has a more significant negative impact on common prosperity in SOEs and companies in the growth life cycle, the government should consider reducing constraints on the operational decision-making of SOEs. Encouraging autonomous decision-making, reducing administrative intervention, and minimizing the negative impact of economic policy uncertainty on them are essential. On the other hand, attention should be given to small and medium-sized growth enterprises. When implementing economic policies, a focus on mitigating the impact on these enterprises is crucial. Negative effects should be minimized or corrected through government support.

## 6.2 Discussion of limitations

**6.2.1 Data limitations.** One of the main data sources used in this study is the China Securities Market and Accounting Research (CSMAR) database. While CSMAR provides comprehensive data on Chinese listed companies, it has several limitations that may affect the robustness of our findings:

1. Non-linear scoring system: The non-linear scoring system used in CSMAR may not be as reliable as linear scoring data. This non-linearity may lead to loss of detailed information, resulting in less precise estimates. Future research could benefit from exploring alternative scoring methods that retain more granular data.

2. Imbalance of shared prosperity indicators: The indicators in the dataset used to measure shared prosperity of enterprises may be somewhat unbalanced. This may affect the accuracy of our measurements on different dimensions. Future research should consider incorporating additional proxy indicators to better capture the development level of these dimensions.

3. Data coverage and update: CSMAR data are updated annually and may not capture more frequent changes in the economic environment or corporate behavior. Researchers should be cautious about the timing of data collection and consider supplementing data sources to ensure more timely analysis.

**6.2.2 Methodological limitations.** This study uses instrumental variable (IV) regression to address endogeneity issues. However, this approach has some methodological limitations:

1. Weak instrumental variables: A key challenge facing IV regression is the possibility of weak instrumental variables. If the correlation between the instrumental variables and the endogenous regressors is weak, the estimates may be biased and the inferences may be unreliable. Future research should adopt more stringent exclusion tests to ensure the validity of the instruments used.

2. Generalizability: The results of this study are based on listed companies in China. Although the generalization has been discussed in this paper, there are still certain limitations, which may limit the generalizability of the results to other environments, such as non-listed companies or companies in different countries. Future research should explore the applicability of these findings to different organizational structures and economic environments.

**6.2.3 Future research directions.** To address these limitations, future research can focus on the following areas:

1. Enhanced data collection: Incorporating more frequent and diverse data sources can improve the accuracy and timeliness of the analysis. This includes exploring alternative databases and integrating real-time data when possible.

2. Robust methodological approach: Using advanced econometric techniques to strengthen the validity of IV regression results, such as using multiple instruments and conducting robustness tests, can provide more reliable insights.

3. Broader contextual analysis: Extending the analysis to non-listed companies, different industries, and other countries would help understand the broader applicability of the findings. Comparative studies in different economic environments could also provide a deeper understanding of the impact of economic policy uncertainty on firm prosperity.

By addressing these limitations, future research could build on the findings of this study and contribute to a more comprehensive understanding of the relationship between economic policy uncertainty and firm prosperity.

## Supporting information

**S1 Data. Manuscript data.**
(XLSX)

## Author Contributions

**Data curation:** Linjing Yang, Xiaoke Tan.

**Formal analysis:** Linjing Yang.

**Methodology:** Guifang Tan.

**Software:** Linjing Yang, Guifang Tan.

**Visualization:** Linjing Yang, Xiaoke Tan.

**Writing – original draft:** Linjing Yang.

**Writing – review & editing:** Linjing Yang.

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
