## [Decision Letter · Decision Letter 0]

9 Jun 2024

PONE-D-24-11742Economic policy uncertainty and common prosperity within the enterprise: Evidence from the Chinese marketPLOS ONE

Dear Dr. Tan,

Thank you for submitting your manuscript to PLOS ONE. After careful consideration, we feel that it has merit but does not fully meet PLOS ONE’s publication criteria as it currently stands. Therefore, we invite you to submit a revised version of the manuscript that addresses the points raised during the review process.

Thank you for your submission and for your patience as we reviewed your manuscript. We appreciate the opportunity to consider your work for publication.

We have carefully reviewed the comments provided by the reviewers, which are attached for your reference, and would like to offer some guidance for revising your manuscript. Please find below a summary of the major points raised by the reviewers and specific suggestions for addressing them:

**Motivation and Contribution:** Reviewer 1 expressed concerns regarding the clarity of the motivation and contribution of your study. To strengthen this aspect, we recommend revisiting the introduction to clearly articulate the significance of your research and its potential implications for both academia and practice. Consider providing a more robust rationale for your study by linking it explicitly to existing literature and theoretical frameworks.**Theoretical Justification:** Reviewer 2 noted the need for a stronger theoretical justification for the relationship between economic policy uncertainty (EPU) and intra-enterprise common prosperity (IECP). As you revise, ensure that your theoretical framework is well-defined and supported by relevant literature. Discuss the underlying economic or finance theories that inform your hypotheses and provide a clear rationale for your research questions.**Methodological Clarity:** Reviewer 3 highlighted several methodological concerns, including the treatment of data, selection of instrumental variables, and clarity in the presentation of the propensity score matching (PSM) test. We recommend providing a more detailed explanation of your methodology, addressing how key variables are defined, measured, and chosen. Clarify the rationale for the choice of instrumental variables and provide a clear description of the procedures used in the PSM test.

We understand that revising your manuscript may require significant effort, but we believe that addressing these points will strengthen your paper and enhance its potential for publication. If you have any questions or need further clarification on any point, please do not hesitate to reach out to us.

Please submit your revised manuscript by Jul 24 2024 11:59PM. If you will need more time than this to complete your revisions, please reply to this message or contact the journal office at plosone@plos.org. Please include the following items when submitting your revised manuscript:A rebuttal letter that responds to each point raised by the academic editor and reviewer(s). You should upload this letter as a separate file labeled 'Response to Reviewers'.A marked-up copy of your manuscript that highlights changes made to the original version. You should upload this as a separate file labeled 'Revised Manuscript with Track Changes'.An unmarked version of your revised paper without tracked changes. You should upload this as a separate file labeled 'Manuscript'.

We look forward to receiving your revised manuscript.

Kind regards,

Pradeep Paraman

Academic Editor

PLOS ONE

Journal Requirements:

Reviewers' comments:

Reviewer's Responses to Questions

**Comments to the Author**

1. Is the manuscript technically sound, and do the data support the conclusions?

Reviewer #1: Partly

Reviewer #2: Partly

Reviewer #3: No

2. Has the statistical analysis been performed appropriately and rigorously? 

Reviewer #1: Yes

Reviewer #2: No

Reviewer #3: No

3. Have the authors made all data underlying the findings in their manuscript fully available?

Reviewer #1: Yes

Reviewer #2: Yes

Reviewer #3: No

4. Is the manuscript presented in an intelligible fashion and written in standard English?

Reviewer #1: Yes

Reviewer #2: Yes

Reviewer #3: Yes

5. Review Comments to the Author

Reviewer #1: 1.the description content about the dependent variable enterprise common prosperity is too little, lack of literature support, please add more literature to explain enterprise common prosperity.

2.The theoretical analysis part of hypothesis 1 does not reflect how economic uncertainty affects prosperity within firms, so please add a theoretical discussion that clarifies the logic and explains how economic uncertainty affects the shared prosperity of firms.

3.Hypotheses 2 and 3 also have the same section as above and do not reflect the logical relationship between the variables in terms of the content section, please reorganise the logic and add more literature to clarify the logical relationship between the variables to make the textual discussion more logical

4.Please explain specifically how the dependent variable, shared prosperity of firms, is measured, which specific indicators are chosen, what is the measurement methodology, and the reasonableness of the scoring.PleasePleasePlease explainPlease explain.Please explain the use ofPlease explain the rationale for using thisPlease explain the methodology used toPlease explain the methodology used to measurePlease explain the methodology used to measure the enterprisePlease explain the methodology used to measure the common prosperity of enterprises.Please explain the methodology used to measure the shared prosperity of enterprises.Please explain the reasonableness of the methodology used to measure the shared prosperity of firms.Please explain the reasonableness of the methodology used to measure the shared prosperity of enterprises.

5.Please explain the rationale for the choice of instrumental variables

6.References in the text are not formatted correctly, please refer to the journal formatting

Reviewer #2: authors should provide revisions that strengthen the theoretical foundation, expand the literature review, enhance methodological rigor, ensure data representativeness, employ rigorous data analysis, offer an in-depth discussion, address potential endogeneity issues.

Reviewer #3: The paper examines the impact of economic policy uncertainty (EPU) on intra-enterprise common prosperity (IECP) using a sample of Chinese listed firms between 2011-2022.

The following comments are listed in the order of the paper’s development.

(1) Both the motivation and the contribution of the paper are weak. The paper motived this research by China’s common prosperity policy. This means that the research has little policy implications for a wider research community. China’s common prosperity policy is only the observed problem for research, which should be raised to the academic level by formulating a research question that can be located to a literature.

(2) Related to the above comment, the paper subjectively linked EPU with IECP without properly discussing the underlying theory to justify this relationship. Although the literature section contains some studies, it is unclear what the underlying economic or finance theory is.

(3) The paper needs to explicitly justify the definition and construction of the IECP score for the firm. What is the economic meaning of the constructed IECP?

(4) In Section 4.2.1, the paper wanted to exclude the effect of the stock market volatility, hence the paper simply deleted the 2015 observations for the sample firms to eliminate the impact of China’s 2015 stock market volatility. Such a treatment implies that the data used in the estimation is no longer a properly panel data set.

(5) In Section 4.3., the paper used ‘the uncertainty of US economic policy’ as an instrument for China’s EPU, but the paper did not explain what ‘the uncertainty of US economic policy’ means, how this variable is defined/constructed, and where the data on this variable is taken from. More importantly, the paper did not even explain why ‘the uncertainty of US economic policy’ is suitable to be an instrumental variable for China’s EPU.

(6) In Section 4.4., the paper undertook a PSM test. However, the motivation of this test is unclear. The explanation about how the paper did the PSM is also unclear. For example, the way in which the paper defines the treatment group and control group is very confusing.

(7) In Section 5.1, the paper did not explain how TFP is constructed. A method is mentioned without the explanation how it is constructed.

(8) The result in column (2) of Table 7 is confusing. In column (2) the estimated coefficient for lnEPU is positively significant in explaining IECP, and the estimated coefficient for the interactive term between lnEPU and productivity (lnlp) is negatively significant in explaining IECP, putting together this result suggests that higher productivity reduces the positive effect of EPU on IECP, which is in the opposite to what the paper wants to prove.

(9) In Table 8, the paper tested the difference in the result between SOEs and non-SOEs. But the paper did not explain how SOEs is defined. Moreover, the paper compared the results between the two groups based on the raw estimates, which is wrong (the same comment applies to both Table 10 and Table 11). In addition, according to the results in Table 8, the IECP of non-SOEs is less affected by EPU, while the negative impact of EPU on the IECP of SOEs is more profound. This result is not consistent with the Chinese practice. The impact of EPU on Chinese non-SOEs should be more profound as compared to that of Chinese SOEs.

6. PLOS authors have the option to publish the peer review history of their article (what does this mean?). If published, this will include your full peer review and any attached files.

Reviewer #1: No

Reviewer #2: No

Reviewer #3: No

---

## [Author Response · Author response to Decision Letter 0]

17 Jul 2024

Thank you to the editor and reviewers for their valuable comments and suggestions on my paper. I have made significant revisions to the paper based on the reviewers’ feedback and corrected typographical errors (important ones include adjusting the data range to 2011-2020 to exclude the impact of the pandemic, as well as some sentence errors).

Reviewer #1:

1.the description content about the dependent variable enterprise common prosperity is too little, lack of literature support, please add more literature to explain enterprise common prosperity.

Thank you for your valuable comments. We have added a new section in the Theoretical Analysis and Research Hypotheses section, which describes the concept of corporate co-prosperity in detail and cites several relevant literature to support our arguments. The new content has been marked in red for easy review. I hope these additions can enhance the theoretical basis and persuasiveness of this paper. Thank you again for your suggestions.

2.The theoretical analysis part of hypothesis 1 does not reflect how economic uncertainty affects prosperity within firms, so please add a theoretical discussion that clarifies the logic and explains how economic uncertainty affects the shared prosperity of firms.

Thank you for your valuable comments. We have re-examined the basic logic of Hypothesis 1, 2, and 3, and added relevant logic and theoretical discussions after each hypothesis to clarify how economic uncertainty affects the common prosperity of enterprises. We have also added more literature support to enhance the clarity of the logical relationship between variables and the logic of the text discussion. I hope these modifications will improve the theoretical and readability of this article. Thank you again for your suggestions.

3.Hypotheses 2 and 3 also have the same section as above and do not reflect the logical relationship between the variables in terms of the content section, please reorganise the logic and add more literature to clarify the logical relationship between the variables to make the textual discussion more logical.

Thank you for your valuable comments. We have re-examined the basic logic of Hypothesis 1, 2, and 3, and added relevant logic and theoretical discussions after each hypothesis to clarify how economic uncertainty affects the common prosperity of enterprises. We have also added more literature support to enhance the clarity of the logical relationship between variables and the logic of the text discussion. I hope these modifications will improve the theoretical and readability of this article. Thank you again for your suggestions.

4.Please explain specifically how the dependent variable, shared prosperity of firms, is measured, which specific indicators are chosen, what is the measurement methodology, and the reasonableness of the scoring.PleasePleasePlease explainPlease explain.Please explain the use ofPlease explain the rationale for using thisPlease explain the methodology used toPlease explain the methodology used to measurePlease explain the methodology used to measure the enterprisePlease explain the methodology used to measure the common prosperity of enterprises.Please explain the methodology used to measure the shared prosperity of enterprises.Please explain the reasonableness of the methodology used to measure the shared prosperity of firms.Please explain the reasonableness of the methodology used to measure the shared prosperity of enterprises.

Thank you for the reviewer's valuable comments. We have explained in detail how the dependent variable is constructed and its rationale, including the specific indicators selected, the measurement method, and the rationale of the score. In addition, the theoretical basis for using this variable is also discussed in the newly added section above (i.e., the section added in response to the first reviewer's comment). The relevant parts have been marked in red for easy review. We hope that these additions will make the measurement method of this article clearer and more convincing.

5.Please explain the rationale for the choice of instrumental variables

Thank you for the reviewer's valuable comments. We have discussed in detail the construction method and reasons for using the instrumental variables, and revised the relevant parts. The main thing is that the instrumental variable can meet the requirements of relevance and exogeneity to ensure its eligibility. I hope these additions can make the selection of instrumental variables in this article more reasonable and convincing.

6.References in the text are not formatted correctly, please refer to the journal formatting

It has been modified according to the journal format.

Reviewer #2: authors should provide revisions that strengthen the theoretical foundation, expand the literature review, enhance methodological rigor, ensure data representativeness, employ rigorous data analysis, offer an in-depth discussion, address potential endogeneity issues。

Thank you for your valuable comments. Based on your suggestions, we have comprehensively sorted out and discussed the article's introduction, literature review, theoretical assumptions, solutions to endogeneity problems, and variable settings. The specific changes are as follows: Strengthen the theoretical basis: Add more relevant literature and theoretical support to the introduction and literature review to strengthen the theoretical basis of the article; Expand the literature review: Expand the literature review section and cite more recent research results to provide more comprehensive background information; Improve the rigor of the method: Detailed explanation of the selection of variables and data processing methods to ensure the rigor of the method; Ensure data representativeness: Detailed discussion of the representativeness of the data to ensure that the data sample is representative; Adopt rigorous data analysis: Adopt more rigorous data analysis methods and describe the analysis process in detail. Solve potential endogeneity problems: Describe in detail the reasons for selecting instrumental variables and how they are constructed to ensure the rationality of the selection, and in-depth explanation of other robustness test methods. The relevant parts have been marked in red.

Reviewer #3:

The following comments are listed in the order of the paper’s development.

(1) Both the motivation and the contribution of the paper are weak. The paper motived this research by China’s common prosperity policy. This means that the research has little policy implications for a wider research community. China’s common prosperity policy is only the observed problem for research, which should be raised to the academic level by formulating a research question that can be located to a literature.

Thank you for the valuable comments from the reviewers. We have reorganized the introduction and the Theoretical Analysis and Research Hypotheses section of the article, and elaborated on the motivation and contribution of the article in more detail. We have expanded the impact of economic policy uncertainty on the common prosperity of enterprises to a more general academic level to enhance the breadth of research and policy influence. The relevant modified parts have been marked in red for easy review. We hope that these changes can enhance the academic value and contribution of this article.

(2) Related to the above comment, the paper subjectively linked EPU with IECP without properly discussing the underlying theory to justify this relationship. Although the literature section contains some studies, it is unclear what the underlying economic or finance theory is.

Thank you for your valuable comments. We have re-examined the theoretical assumptions on the impact of economic policy uncertainty (EPU) on intra-enterprise common prosperity (IECP) and clarified the logical relationship and theoretical basis of the relationship between EPU and IECP. We have discussed the relevant economic and financial theories in detail in the article to support the rationality and academic nature of this relationship. The relevant revisions have been marked in red for easy review.

(3) The paper needs to explicitly justify the definition and construction of the IECP score for the firm. What is the economic meaning of the constructed IECP?

Thank you for the reviewer's valuable comments. We have explained in detail the definition and rationality of the construction of the company's IECP score, including the specific construction method and theoretical basis. In addition, we discussed the economic significance of IECP in the newly added subsection 2.1 "The connotation of common prosperity within the enterprise". The relevant modified parts have been marked in red for easy review. We hope that these additions will make the measurement method and economic significance of this article clearer and more convincing.

(4) In Section 4.2.1, the paper wanted to exclude the effect of the stock market volatility, hence the paper simply deleted the 2015 observations for the sample firms to eliminate the impact of China’s 2015 stock market volatility. Such a treatment implies that the data used in the estimation is no longer a properly panel data set.

Thank you for your valuable comments. We deleted the 2015 data in the robustness test in order to verify the adaptability of the data in different situations. After deleting the 2015 data, the conclusion of the article did not change, which just shows that our data conclusion is robust and reliable and can be used for subsequent research. Therefore, this treatment does not affect the appropriateness of the data set, but enhances the credibility of the conclusion. I hope these explanations can clarify your doubts. Thank you again for your suggestions.

(5) In Section 4.3., the paper used ‘the uncertainty of US economic policy’ as an instrument for China’s EPU, but the paper did not explain what ‘the uncertainty of US economic policy’ means, how this variable is defined/constructed, and where the data on this variable is taken from. More importantly, the paper did not even explain why ‘the uncertainty of US economic policy’ is suitable to be an instrumental variable for China’s EPU.

Thank you for the reviewer's valuable comments. We have already described in detail the definition, construction method and data source of the instrumental variable "U.S. economic policy uncertainty" in the original paper. In addition, we have also explained in detail why "U.S. economic policy uncertainty" is suitable as an instrumental variable for China's EPU. The relevant parts have been marked in red for easy review. I hope these additions can make the choice of instrumental variables in this article clearer and more convincing.

(6) In Section 4.4., the paper undertook a PSM test. However, the motivation of this test is unclear. The explanation about how the paper did the PSM is also unclear. For example, the way in which the paper defines the treatment group and control group is very confusing.

Thank you for the valuable comments from the reviewers. Considering the refinement and usefulness of the article, we carefully evaluated the contribution of the PSM test to the robustness test of the article and decided to delete the relevant parts. This can remove redundant content that contributes little to the article and enhance the refinement and readability of the article. We hope that these modifications can improve the overall quality of this article.

(7) In Section 5.1, the paper did not explain how TFP is constructed. A method is mentioned without the explanation how it is constructed.

Thank you for the reviewer's valuable comments. We have described in detail how TFP (total factor productivity) is constructed in Section 5.1 and marked the relevant parts in red for easier review. We hope that these additions will make the measurement method of this paper clearer and more convincing.

(8) The result in column (2) of Table 7 is confusing. In column (2) the estimated coefficient for lnEPU is positively significant in explaining IECP, and the estimated coefficient for the interactive term between lnEPU and productivity (lnlp) is negatively significant in explaining IECP, putting together this result suggests that higher productivity reduces the positive effect of EPU on IECP, which is in the opposite to what the paper wants to prove.

Thanks to the reviewers for their valuable comments. We have explained in detail how productivity is built and conducted an in-depth analysis of the moderating effects. Based on the basic regression, we gradually added moderator variables and interaction terms.

In column (1), we add total factor productivity (TFP) and find that it significantly changes the sign of the explanatory variable (from negative to positive), which indicates that TFP may be an important mechanism variable that can improve economic policy uncertainty. The negative relationship between sex (EPU) and intra-firm shared prosperity (IECP). Therefore, in column (2), we add the interaction term of the explanatory variable and the moderator variable, and the results show that it significantly alleviates the negative relationship in the base regression, which is consistent with our expectations.

Regarding the change in sign of the explanatory variables, the addition of moderator variables and interaction terms requires re-estimation of the negative effects of the original basic regression. The added model already considers the impact of the moderator variable on the explanatory variable at different levels, so in a model with moderator variables, the estimated value of the explanatory variable does not need to be considered too much. The results of the interaction term in this article have demonstrated the establishment of the core hypothesis.

Hopefully these explanations will clear up your doubts. Thanks again for your advice.

(9) In Table 8, the paper tested the difference in the result between SOEs and non-SOEs. But the paper did not explain how SOEs is defined. Moreover, the paper compared the results between the two groups based on the raw estimates, which is wrong (the same comment applies to both Table 10 and Table 11). In addition, according to the results in Table 8, the IECP of non-SOEs is less affected by EPU, while the negative impact of EPU on the IECP of SOEs is more profound. This result is not consistent with the Chinese practice. The impact of EPU on Chinese non-SOEs should be more profound as compared to that of Chinese SOEs.

Thank you for the reviewer's valuable comments. We have explained the definition of state-owned enterprises in detail in the article. In addition, considering the drawbacks of comparing the two groups of results directly based on the original estimates, we used the inter-group difference coefficient to compensate for it on this basis to ensure that the differences between the groups can be accurately compared.

Regarding the reviewer's point that the impact of EPU on China's non-state-owned enterprises should be more far-reaching, we believe that state-owned enterprises have a stronger policy orientation than non-state-owned enterprises. When the country's macroeconomic policies change, state-owned enterprises have less flexibility and can only accept government regulation more directly, which also determines that the uncertainty of economic policy changes will have a greater impact on state-owned enterprises. Therefore, it is reasonable that our results show that the negative impact of EPU on the IECP of state-owned enterprises is more far-reaching.

I hope these explanations can clarify your doubts.

---

## [Editor Report · Decision Letter 1]

22 Jul 2024

PONE-D-24-11742R1Economic policy uncertainty and common prosperity within the enterprise: Evidence from the Chinese marketPLOS ONE

Dear Dr. Tan,

Thank you for submitting your manuscript to PLOS ONE. After careful consideration, we feel that it has merit but does not fully meet PLOS ONE’s publication criteria as it currently stands. I have now completed a thorough review of your paper. I am pleased to inform you that your manuscript shows promise but requires revisions before it can be considered for publication. Therefore, we invite you to submit a revised version of the manuscript that addresses the points raised during the review process.

**Specific Areas for Improvement:**

<ol><li>**Clarity and Structure:**

Please revise the introduction (Section 1) to provide clearer contextualization of the study's objectives and the significance of the research question addressed. Consider succinctly summarizing the theoretical framework and key hypotheses.In Section 3.2.1 (Dependent Variable), ensure that the definition and operationalization of "common prosperity" are clearly articulated. Specify the exact methodology used for constructing the enterprise common prosperity score.Improve the flow of the methodology section (Section 4.3) by clearly delineating between the first and second stages of the instrumental variable regression. Additionally, provide a more detailed explanation of the rationale behind choosing US economic policy uncertainty as the instrumental variable.<li>**Contextualization and Generalizability:**

Expand the discussion in Section 5 on the implications of your findings beyond the specific context of listed companies in China. Discuss how the results might apply to other sectors or countries facing similar issues of economic policy uncertainty and corporate social responsibility.Consider adding a subsection under Section 5.2.1 (Equity Nature Perspective) to explicitly discuss the potential limitations and biases associated with using state-owned enterprises (SOEs) and non-state-owned enterprises (non-SOEs) as distinct groups. Address the generalizability of findings to different organizational structures.<li>**Discussion of Limitations:**

Provide a more comprehensive discussion of potential limitations associated with the data sources (e.g., CSMAR) and the methodology used (e.g., IV regression). Discuss how these limitations might affect the interpretation of your results and suggest avenues for future research to address these limitations.<li>**Interpretation of Results:**

Strengthen the discussion in Section 5.1 (Moderation Effect Test) by offering deeper insights into the practical implications of your findings for policymakers and practitioners interested in promoting common prosperity within enterprises.

**Reviewer’s Focus Areas:**

The reviewers have highlighted the need for improved clarity in presenting methods and results, particularly in Sections 3.2.1 and 4.3.They also emphasized the importance of discussing the broader implications and limitations of your study findings, as outlined above.

**Next Steps:** Please address these specific comments and revise your manuscript accordingly. Once you have made these revisions, we would be delighted to reconsider your manuscript for potential publication in our journal.

Thank you once again for choosing to submit your work to us. Please submit your revised manuscript by Sep 05 2024 11:59PM. If you will need more time than this to complete your revisions, please reply to this message or contact the journal office at plosone@plos.org. Please include the following items when submitting your revised manuscript:A rebuttal letter that responds to each point raised by the academic editor and reviewer(s). You should upload this letter as a separate file labeled 'Response to Reviewers'.A marked-up copy of your manuscript that highlights changes made to the original version. You should upload this as a separate file labeled 'Revised Manuscript with Track Changes'.An unmarked version of your revised paper without tracked changes. You should upload this as a separate file labeled 'Manuscript'.If applicable, we recommend that you deposit your laboratory protocols in protocols.io to enhance the reproducibility of your results. Protocols.io assigns your protocol its own identifier (DOI) so that it can be cited independently in the future. For instructions see: https://journals.plos.org/plosone/s/submission-guidelines#loc-laboratory-protocols. Additionally, PLOS ONE offers an option for publishing peer-reviewed Lab Protocol articles, which describe protocols hosted on protocols.io. Read more information on sharing protocols at https://plos.org/protocols?utm_medium=editorial-email&utm_source=authorletters&utm_campaign=protocols.

We look forward to receiving your revised manuscript.

Kind regards,

Pradeep Paraman

Academic Editor

PLOS ONE
---

## [Author Response · Author response to Decision Letter 1]

1 Aug 2024

First of all, I would like to thank the reviewers and editors for their valuable suggestions. We have improved the four areas you mentioned, taking into account clarity, broad applicability, and limitations. The following is my response to the review comments.

Clarity and Structure:

1、Please revise the introduction (Section 1) to provide clearer contextualization of the study's objectives and the significance of the research question addressed. Consider succinctly summarizing the theoretical framework and key hypotheses.

Thank you for the reviewer's valuable comments. We have reorganized the introduction, clarified the research objectives, and highlighted the importance of the research questions. In addition, in Section 2, "Theoretical Analysis and Research Hypotheses", we briefly summarize the theoretical framework and key hypotheses through a conceptual framework diagram. The relevant parts have been marked in red.

2、In Section 3.2.1 (Dependent Variable), ensure that the definition and operationalization of "common prosperity" are clearly articulated. Specify the exact methodology used for constructing the enterprise common prosperity score.

This article has clarified the definition and construction method of common prosperity, and announced the details of the selection of three-level indicator variables. The variable construction method of this article is also clear, mainly reverse indicator adjustment, missing value filling, standardization and weighting to obtain the common prosperity score within the enterprise, and finally divide it into 9 levels. The relevant parts are marked in red.

3、Improve the flow of the methodology section (Section 4.3) by clearly delineating between the first and second stages of the instrumental variable regression. Additionally, provide a more detailed explanation of the rationale behind choosing US economic policy uncertainty as the instrumental variable.

This paper clearly divides the first and second stages of instrumental variable regression and improves the method flow. In addition, this paper elaborates on the reasons for selecting US economic policy uncertainty as an instrumental variable from the perspective of correlation and exogeneity. The relevant parts are marked in red.

Contextualization and Generalizability:

4、Expand the discussion in Section 5 on the implications of your findings beyond the specific context of listed companies in China. Discuss how the results might apply to other sectors or countries facing similar issues of economic policy uncertainty and corporate social responsibility.

Thanks to the reviewers for their valuable comments. We have expanded the discussion in Section 5 to explore the applicability of economic policy uncertainty, corporate social responsibility, total factor productivity and internal prosperity in different industries and countries from an international and industry perspective. The relevant parts are marked in red.

5、Consider adding a subsection under Section 5.2.1 (Equity Nature Perspective) to explicitly discuss the potential limitations and biases associated with using state-owned enterprises (SOEs) and non-state-owned enterprises (non-SOEs) as distinct groups. Address the generalizability of findings to different organizational structures.

Thank you for the reviewer's valuable comments. We have discussed the generalizability of different organizational structures in Section 5.2.1 and added a new Section 5.2.2 "Potential limitations and biases" to discuss in detail the limitations and biases of using state-owned enterprises (SOEs) and non-SOEs as different groups. The relevant parts have been marked in red.

Discussion of Limitations:

6、Provide a more comprehensive discussion of potential limitations associated with the data sources (e.g., CSMAR) and the methodology used (e.g., IV regression). Discuss how these limitations might affect the interpretation of your results and suggest avenues for future research to address these limitations.

Thank you for your valuable comments. We have added a new section 6.2 "Discussion of Limitations" under the conclusion of Section 6, which elaborates on the potential limitations of the article from the two aspects of data limitations and method limitations, and proposes solutions and directions to these limitations in section 6.2.3 "Future Research Directions". The specific changes have been marked in red in the original text. I hope these changes can better meet the requirements of the reviewers and improve the overall quality of the paper. Thank you again for the reviewers' valuable suggestions.

Interpretation of Results:

7、Strengthen the discussion in Section 5.1 (Moderation Effect Test) by offering deeper insights into the practical implications of your findings for policymakers and practitioners interested in promoting common prosperity within enterprises.

Thanks to the reviewers’ comments, we have strengthened the discussion in Section 5.1.

---

## [Editor Report · Decision Letter 2]

12 Aug 2024

Economic policy uncertainty and common prosperity within the enterprise: Evidence from the Chinese market

PONE-D-24-11742R2

Dear Dr. Tan,

We’re pleased to inform you that your manuscript has been judged scientifically suitable for publication and will be formally accepted for publication once it meets all outstanding technical requirements.

Kind regards,

Pradeep Paraman

Academic Editor

PLOS ONE
---

## [Editor Report · Acceptance letter]

22 Aug 2024

PONE-D-24-11742R2 

PLOS ONE

Dear Dr. Tan, 

I'm pleased to inform you that your manuscript has been deemed suitable for publication in PLOS ONE. Congratulations! Your manuscript is now being handed over to our production team.

Kind regards, 

on behalf of

Dr. Pradeep Paraman 

Academic Editor

PLOS ONE